# Factors Affecting Nursing Surveillance Activity among Clinical Nurses

**DOI:** 10.3390/healthcare11091273

**Published:** 2023-04-29

**Authors:** Se Young Kim, Mi-Kyoung Cho

**Affiliations:** 1Department of Nursing, Changwon National University, 20 Changwondaehak-ro, Uichang-gu, Changwon 51140, Republic of Korea; 2Department of Nursing Science, Chungbuk National University, 1 Chungdae-ro, Seowon-gu, Cheongju 28644, Republic of Korea

**Keywords:** safety management, nursing, patient safety, professional competence

## Abstract

Nursing surveillance is a defense mechanism that protects patients from adverse events, as it is a systematic process that contributes to nurses’ detection of and intervention in dangerous situations. This descriptive cross-sectional study examined the degree of nursing surveillance activity performed by clinical nurses, nurses’ perceived importance of nursing surveillance, and their perception of patient safety culture. The study aimed to identify the predictors of nursing surveillance activity. Participants included 205 clinical nurses from two secondary hospitals and two tertiary hospitals in Changwon-si and Cheongju-si, South Korea, who had at least one year of clinical experience. Nursing surveillance activity was high among nurses who were assigned fewer than 1.88 patients, who worked in a tertiary hospital, and those who scored 7.0 or higher in nurses’ perceived expertise. Nursing surveillance activity was significantly positively correlated with nurses’ perceived importance of nursing surveillance and patient safety culture. A hierarchical multiple regression analysis was performed to identify the predictors of nursing surveillance activity. Nurses’ perceived expertise, perceived importance of nursing surveillance, patient safety culture, and type of hospital explained 65.3% of the variance of nursing surveillance activity. This study is significant as it provides an assessment of the extent and key predictors of nursing surveillance activity.

## 1. Introduction

The coronavirus disease 2019 (COVID-19) pandemic has provided us with a good example of surveillance—how data are collected, interpreted, and synthesized to aid in the decision-making process to promote safety [1]. The Institute of Medicine (IOM) suggests that nurses are the best-positioned healthcare providers for preventing complications, detecting risks, and appropriately responding to several healthcare situations [2]. Nursing surveillance is the process of collecting, reviewing, interpreting, and analyzing data to identify and prevent potential complications [3].

The Nursing Interventions Classification (NIC) defines nursing surveillance as the “purposeful and ongoing acquisition, interpretation, and synthesis of patient data for clinical decision making” [4]. Nurses interact with patients and their families and continually collect and evaluate patient data, which include physical assessment findings, physiological indices on monitors, the effectiveness of the medication, patients’ responses to interventions, and laboratory and diagnostic test results [3]. The key to nursing surveillance is to recognize patterns in data and identify their relevance and similarities to classify patients’ risk [5]. Ultimately, nursing surveillance reduces the incidence of adverse events and contributes to patients’ safety [6].

According to the Eindhoven Model, nursing surveillance is a defense mechanism that protects patients from adverse events [7] and systematically contributes to nurses’ detection of and intervention in dangerous situations [8]. Although increasing nursing surveillance interventions often results in the hospital incurring additional costs, a past study showed that patients who received nursing surveillance at least 12 times a day had a significantly lower rate of falls, which led to a reduction in the treatment costs for falls [9]. Further, patients whose vital signs were frequently monitored according to nursing diagnoses related to nursing surveillance had a higher rate of survival from cardiac arrest [3].

Organizational culture was proposed as an antecedent to nursing surveillance [6]. Particularly, the organizational culture of hospitals influences patient safety culture and fosters better safety practices within the organization [10,11]. The IOM stressed the importance of cultivating a patient safety culture that protects patients from harm [12]. A patient safety culture encompasses communication based on mutual respect, an unhindered exchange of information, shared awareness of the importance of safety, organizational learning, leadership, non-punitive thinking, and error-reporting protocols [13]. Patient safety culture is negatively correlated with the incidence of adverse events [14,15,16,17], and “learning climate”—a domain of patient safety culture—decreased medication errors by nurses [18]. Wang et al. examined the relationship between patient safety culture and adverse events, reporting that “manager support for patient safety” and “frequency of events reported”—both of which are domains of patient safety culture—lowered medication errors, while “organizational learning and continuous improvement”, “feedback and communication about the error”, “non-punitive response to error”, and “frequency of events reported” lowered the incidence of pressure ulcers [19].

Patient safety culture had positive effects on nursing activity related to the identification, resolution, and prevention of problems that occurred or may occur during patient care in hospitals [20,21]. Certain domains of patient safety culture—“manager’s attitude” and “frequency of events reported”—influenced safe nursing practice [22]. In this context, nurses’ perceived patient safety culture is predicted to affect nursing surveillance activity, which involves classifying patient risks and halting dangerous situations [5,6].

Halverson and Tilley conceptualized nursing surveillance in their literature review and presented nurses’ education and expertise, nursing staffing, organizational culture, and work environment as some antecedents to nursing surveillance [7]. Kutney-Lee et al. defined nursing surveillance capacity as an organization’s efforts to reinforce or reduce nursing surveillance and reported that the nurse-to-patient ratio, nurses’ level of education, and an excellent nursing work environment are associated with a low rate of adverse events [23]. In recent years, studies have identified nurses’ level of education, knowledge, experience, training, and skills [24,25,26]; staffing level; the presence of a culture supportive of nursing surveillance; and interdisciplinary communication [6,27] as some predictors of nursing surveillance. Although studies, such as Kelly [28], did explore the multidimensionality of nursing surveillance using validated interventions from the NIC, studies that investigate the extent of nursing surveillance interventions performed by nurses in hospitals are scarce. Further, research on the correlation between nursing surveillance and nurses’ characteristics, staffing level, and hospital patient safety culture has been lacking [29]. In this study, nursing surveillance uses the definition of purposeful and ongoing acquisition, interpretation, and synthesis of patient data for clinical decision-making, and nursing surveillance activity was measured by the activities of surveillance in NIC [4]. Therefore, this study examined the degree of nursing surveillance activity performed by clinical nurses in practice, nurses’ perceived importance of nursing surveillance, and their perception of patient safety culture to identify the predictors of nursing surveillance activity.

## 2. Materials and Methods

### 2.1. Study Design

This study is a descriptive cross-sectional survey that investigates the association between clinical nurses’ surveillance activity and their perceived importance of nursing surveillance and patient safety culture.

### 2.2. Participants

A total of 205 nurses with at least one year of clinical experience were recruited from two secondary hospitals and two tertiary hospitals in Changwon-si and Cheongju-si, South Korea, via a voluntary online survey that explained the study purpose and methods. The sample size was determined using G*power version 3.1.9.4 (Heinrich-Heine-Universität, Düsseldorf, Germany) [30]. To achieve a significance level (α) of 0.05 for a two-tailed test with a medium effect size (f^2^) of 0.15, as suggested by Cohen [31] for hierarchical linear multiple regression analysis, power (1-β) of 0.95, and eight predictors in Model 1 with 21 predictors, the necessary sample size was calculated to be 161. Considering a possible 20% withdrawal rate, the target sample size was set to 202. All of the recruited participants completed the survey during the data collection period. All 205 questionnaires were included in the final analysis, and the final power of the study was close to 1.00.

### 2.3. Tools

#### 2.3.1. Nursing Surveillance Activity and Perceived Importance of Nursing Surveillance

To investigate the frequency of nurses’ nursing surveillance activity, we incorporated 46 of the surveillance interventions stated in the NIC. These were translated into Korean by a professional translator (forward translation) after we obtained permission from both the authors and publisher of the NIC to proceed with the Korean translation and to use the activities in a questionnaire [32]. The 46 nursing surveillance activities were developed into questionnaire items, and their content validity (content validity index; CVI) was evaluated by an expert panel consisting of two nursing professors, one nurse manager, and one nursing Ph.D. student with nurse management experience. The items’ relevance to clinical nursing surveillance activity and the appropriateness of the Korean translations concerning expression and vocabulary were evaluated on a four-point Likert scale: 1 = “not relevant”, 2 = “somewhat relevant”, 3 = “quite relevant”, and 4 = “very relevant”. The experts were asked to comment and provide feedback for revision if they marked an item as “not relevant” or “somewhat relevant”. The item-content validity index (I-CVI) was calculated according to the ratings from four reviewers. Thirteen items with an I-CVI of 0.80 or lower were revised for terminology and context based on the experts’ comments, after which we conducted a second round of content validity testing. The second round of content validity testing was performed by a panel of experts comprising two nursing management professors and two charge nurses. The CVI for all items was 1; thus, all items were included. The modified Korean items were back-translated by another professor translator. A nursing professor who obtained a Ph.D. degree from a university in the United States and has teaching experience compared and reviewed the semantic similarity of the surveillance activities in the NIC and the back-translated items, and the translation’s validity was evaluated by the panel of experts.

The final questionnaire contained 46 items that were relevant to clinical nurses providing patient care. Each item was rated on a five-point Likert scale from 0 = “never” to 4 = “Always”, with a higher score indicating a higher degree of nursing surveillance activity. The reliability of the nursing surveillance activity scale, as measured with Cronbach’s alpha, was 0.961. Nurses’ perceived importance of nursing surveillance was rated on a five-point Likert scale from 0 = “not important at all” to 4 = “very important”, with a higher score indicating a greater perceived importance of the interventions. The reliability of the perceived importance of the nursing surveillance scale, as measured with Cronbach’s alpha, was 0.975.

#### 2.3.2. Patient Safety Culture

The Korean version [33] of the Hospital Survey on Patient Safety Culture TM (SOPS^TM^) Version 2.0, originally developed by the US Agency for Healthcare Research and Quality (https://www.ahrq.gov/sops/surveys/hospital/index.html accessed on 28 December 2022), was used after obtaining permission from the authors. The 33-item SOPS comprises 10 composite measures and two single-item measures: teamwork (3 items); staffing and work pace (3 items); organizational learning and continuous improvement (3 items); response to error (4 items); supervisor, manager, or clinical leader support for patient safety (3 items); communication about the error (3 items); openness in communication (4 items); reporting patient safety events (2 items); hospital management support for patient safety (3 items); handoffs and information exchange (3 items); the number of events reported; and patient safety rating. Each item is rated on a five-point Likert scale ranging from 1 = “strongly disagree” to 5 = “strongly agree”. There were twelve reverse-scored items, in which a higher score indicates a more positively perceived patient safety culture. In Lee and Dahinten, the reliability of 9 out of 10 composite measures (Cronbach’s alpha) was 0.70 or higher, and that for staffing and work pace was 0.61 [33]. In this study, Cronbach’s alpha for the 10 composite measures ranged from 0.645 to 0.833, and Cronbach’s alpha for all 33 items was 0.860.

#### 2.3.3. Participants’ Characteristics

Eight participant characteristics were surveyed: age, sex, total work experience (in years), department, nurse-to-patient ratio, method of nursing care delivery, type of hospital, and nurses’ perceived expertise. For total work experience, the number of months worked was converted into years, and the nurse-to-patient ratio was calculated according to the number of beds and nurses in a unit. Nurses’ perceived expertise was rated on a five-point Likert scale ranging from 1 = “strongly disagree” to 5 = “strongly agree”.

### 2.4. Data Collection and Ethical Considerations

After obtaining approval from the Institutional Review Board at the authors’ affiliation (7001066-202209-HR-059), we visited the nursing departments of the study hospitals to explain the purpose and methods of the study and obtain permission and cooperation to collect data from medical-surgical and intensive care units’ nurses with at least one year of clinical experience. Data were collected from 4 to 22 January 2023, and the nursing department distributed an announcement containing the QR code and link to the online survey. The announcement specified the study purpose, contents, and procedures and also provided information on participant anonymity, freedom to withdraw from the study at any time, and the researchers’ contact information (phone number, email address) for any questions. The online questionnaire was structured so that only nurses who clicked “agree” to express consent to participate could proceed with the survey. Online consent by clicking “agree” was accepted in place of a physical form as this study was exempted from the requirement to obtain written consent forms. Participants’ cell phone numbers were collected so we could send them a mobile coupon after completing the survey.

### 2.5. Statistical Analysis

The collected data were analyzed using SPSS 26.0 (IBM, Armonk, NY, USA). Participant characteristics, nursing surveillance activity and perceived importance of nursing surveillance, and patient safety culture were analyzed with descriptive statistics, including real numbers and percentages, mean with standard reference, and minimum and maximum values. Differences in nursing surveillance activity according to participants’ characteristics were analyzed with independent *t*-tests. The correlations among nursing surveillance activity, perceived importance of nursing surveillance, and patient safety culture were analyzed with Pearson’s correlation coefficients. The predictors of nursing surveillance activity were analyzed with modeling variables that were significant in the univariate analysis. A hierarchical linear multiple regression analysis was performed: Model 1 contained participant characteristics, Model 2 contained patient safety culture domains, and Model 3 contained nurses’ perceived importance of nursing surveillance. To ensure a parsimonious model, variables were entered via a stepwise method. The assumptions of multiple regression analysis were tested with residual plots, standardized residuals, tolerance, variance inflation factor (VIF), and Durbin–Watson values. Significance was set to <0.05.

## 3. Results

### 3.1. Participants’ Characteristics

Participants’ characteristics are shown in Table 1.

### 3.2. Descriptive Statistics of the Variables

The mean nursing surveillance activity score was 149.57 ± 23.39 (range: 85–184), the mean perceived importance of nursing surveillance was 156.94 ± 21.57 (range: 92–184), and the mean patient safety culture score was 112.47 ± 15.16 (range: 48–165) (Table 2).

### 3.3. Nursing Surveillance Activity According to Participants’ Characteristics

Nursing surveillance activity was higher among nurses assigned fewer than 1.88 patients than among those assigned 1.88 patients or more (*t* = 2.43, *p* = 0.016). Nursing surveillance activity was higher among nurses working in a tertiary hospital than among those working in a secondary hospital (*t* = 2.59, *p* = 0.010). Nursing surveillance activity was higher among those who scored 7.0 or higher than among those who scored below 7.0 in nurses’ perceived expertise (*t* = 3.24, *p* = 0.001). No significant differences were observed in nursing surveillance activity according to age, sex, work experience, department, and the method of nursing care delivery (Table 3).

### 3.4. Correlation among Variables

Nursing surveillance activity was significantly positively correlated with nurses’ perceived importance of nursing surveillance (*r* = 0.80, *p* < 0.001) and patient safety culture (*r* = 0.26, *p* < 0.001). Nursing surveillance activity was also significantly positively correlated with multiple domains of patient safety culture: teamwork (*r* = 0.22, *p* = 0.001), organizational learning and continuous improvement (*r* = 0.29, *p* < 0.001), error communication (*r* = 0.26, *p* < 0.001), openness in communication (*r* = 0.28, *p* < 0.001), patient safety incident reporting (*r* = 0.19, *p* = 0.007), and experience in patient safety incident reporting (*r* = 0.28, *p* < 0.001; Table 4).

### 3.5. Factors Affecting Nursing Surveillance Activity

A hierarchical regression analysis was performed to identify the predictors of nursing surveillance activity; the regression model is presented in Table 5. The categorical variables that were significant in the univariate analysis, such as nurse-to-patient ratio and type of hospital, were dummy-coded, and the nurse’s perceived expertise was entered as a continuous variable. These variables were used to establish Model 1 using stepwise multiple regression analysis. Model 2 was established using stepwise multiple regression analysis by entering the patient safety culture domains that were significant in the univariate analysis: teamwork, organizational learning and continuous improvement, error communication, openness in communication, patient safety incident reporting, and experience in patient safety incident reporting. Finally, Model 3 was created by entering the perceived importance of nursing surveillance. The variables to be entered in the regression models were selected according to an alpha of 0.05, and variables were removed according to an alpha of 0.1. In the nursing surveillance activity model, tolerance between independent variables was above the cutoff of 0.1, and VIF met the cutoff of ≤2. The Durbin–Watson statistic was close to 2, confirming the absence of multicollinearity.

In Model 1, which includes participants’ characteristics, nurses’ perceived expertise (*t* = 3.24, *p* = 0.001) and tertiary hospital with reference to the secondary hospital (*t* = 2.59, *p* = 0.010) were identified as significant predictors. With these two predictors, Model 1 accounted for 7.1% of the variance of nursing surveillance activity (F = 8.75, *p* < 0.001).

In Model 2, which contains the patient safety culture domains, organizational learning and continuous improvement (*t* = 2.79, *p* = 0.006) and openness in communication (*t* = 3.00, *p* = 0.003) were identified as significant predictors. This two-factor Model 2, which is a hierarchical regression model of Model 1, explained 15.5% of the variance of nursing surveillance activity (F = 10.33, *p* < 0.001).

In Model 3, which was created using the hierarchical entry of perceived importance of nursing surveillance from Models 1 and 2, nurses’ perceived importance of nursing surveillance was identified as the significant predictor (*t* = 16.96, *p* < 0.001). This one-factor regression model explained 65.3% of the variance of nursing surveillance activity (F = 77.65, *p* < 0.001).

## 4. Discussion

Nursing surveillance encompasses cognitive and behavioral factors and is the process of interventions administered by nurses based on their monitoring, assessment, and patient status [3]. This study investigated the level of nursing surveillance activities, representing the behavioral factors of nursing surveillance performed by nurses in acute care hospitals. We examined the effects of nurses’ characteristics and work environment, particularly patient safety culture, on nursing surveillance activity.

In this study, the mean nursing surveillance activity score was 3.25 out of 4, suggesting that nurses frequently performed the nursing surveillance interventions outlined in the NIC. This was similar to the results reported by Kelly [28], in which nurses claimed to frequently perform most of the interventions. Furthermore, Model 3, which incorporates nurse characteristics, patient safety culture, and nurses’ perceived importance of nursing surveillance, showed that nursing surveillance activity increased when the perceived importance of nursing surveillance, which accounted for 49% of all nursing surveillance activity, increased. Similarly, studies that investigated nurses’ practice and perception of infection control using the same instruments showed that nurse perception is the most potent predictor of infection control practices, highlighting the need for periodic and systematic education [34,35]. Particularly, practical training, such as simulation-based education, can better enhance nurses’ clinical performance compared with lectures that solely focus on imparting knowledge. Therefore, simulation-based education programs that incorporate both theory and practicum need to be developed and implemented [34]. In this context, to promote nursing surveillance activity among nurses in acute care hospitals, simulation-based education programs that emphasize the importance of nursing surveillance and train nurses to detect changes in patient status, synthesize data to make clinical judgments, and provide appropriate interventions are required.

In this study, Model 2, which included nurses’ characteristics and patient safety culture as the independent variables, showed that nursing surveillance activity increased when increasing “organizational learning and continuous improvement” and “openness in communication”, domains of patient safety culture, along with nurses’ perceived expertise. These factors explain 15.5% of nursing surveillance activity. These results are similar to previous findings showing that various domains of patient safety culture positively contributed to safe nursing practice in which nurses prevented and ameliorated patients’ problems [20,21,22,36]. Patient safety culture is the individual and organizational behavior patterns based on shared beliefs and values aimed at minimizing harm to patients that could occur during the delivery of healthcare [37]. Patient safety culture influences the attitudes and behaviors pertinent to the compliance of safety regulations among members of an organization [38]. Regarding the antecedents of nursing surveillance, Peet et al. [29] argued that bedside nursing competency, which is fostered by nurses’ collaboration based on patient safety culture, is crucial; and Halverson and Tilley [27] argued that a culture supportive of nursing surveillance must be cultivated. Although several studies have presented the properties and predictors of nursing surveillance activity [3,6,8,39], studies rarely investigated the association between organizational culture and nursing surveillance. Therefore, this study is significant in that it sheds light on the importance of patient safety culture as an antecedent of nursing surveillance. Our findings highlight the importance of organizational strategies and how they can promote nurses’ surveillance activity. More specifically, hospitals can encourage quality improvement activities and organizational learning to promote patient safety and increase openness in communication so that nurses can monitor patients’ status and notify physicians to respond to the situation effectively.

Model 1, which included nurses’ characteristics and type of hospital as independent variables, showed that nurses’ subjective expertise positively affected nursing surveillance activity, and expertise remained a significant predictor even after entering patient safety culture and nurses’ perceived importance of nursing surveillance. These results support the argument from previous studies that nurses’ expertise is the most powerful predictor of nursing surveillance [23,39,40,41,42]. While new graduate nurses generally fail to see beyond individual components of a work process and recognize the underlying pattern, more experienced nurses comprehend the entirety of a situation, discern patterns that may emerge, identify potential issues, and implement appropriate interventions [39]. Seasoned nurses not only follow written prescription orders but also utilize patient history and clinical information to assess patients and their families from a broader perspective, and they ask more questions to obtain accurate information during handoff [6,43]. In nursing, expertise refers to the integration of subjective knowledge and experience [39,44]. Therefore, to strengthen nurses’ expertise, it is necessary to provide specialized education and training in nursing surveillance and to implement an early warning system, checklists, and clinical information system to support nurses lacking experience [6].

Tertiary hospital nurses performed more nursing surveillance interventions than general hospital nurses, and the group with a nurse-to-patient ratio of ≤1.88 had a significantly higher nursing surveillance activity score than the group with a nurse-to-patient ratio of ≥1.88. These results may be attributable to the fact that tertiary hospitals, which are larger hospitals that provide more specialized care than general hospitals, have more high-acuity patients and lower nurse-to-patient ratios, thereby contributing to a greater degree of nursing surveillance activity. Nurse staffing is an essential antecedent of nursing surveillance, and adequate staffing is an environmental factor that facilitates nursing surveillance [26]. The current results highlight the need for hospitals to ensure adequate nurse staffing to promote nursing surveillance, which contributes to patient safety.

The cross-sectional nature of this study hinders a causal analysis among the variables, and the study has limitations in generalizing the results because the survey was conducted on nurses from two general hospitals and two tertiary hospitals in two cities in South Korea, selected by convenience sampling. In the future, studies should conduct multi-center studies to investigate nursing surveillance activity among different types of nursing delivery systems and analyze the associations of nursing surveillance with the factors proposed as outcomes of nursing surveillance. Additionally, given that the nursing surveillance interventions listed on the NIC were used in this study, we recommend that subsequent studies conceptualize nursing surveillance from clinical nurses’ perspectives and explore their lived experiences to develop and validate instruments that capture the multidimensional nature of nursing surveillance.

## 5. Conclusions

Nursing surveillance around the clock is crucial to patients’ health and safety. This study found that nurses’ characteristics and work environment can be proposed as antecedents of nursing surveillance [3,6,8,39], specifically, nurses’ expertise, their perception of nursing surveillance, patient safety culture, and type of hospital-predicted nursing surveillance activity. In order to promote nurses’ surveillance activities in acute hospitals, it is necessary to maintain an appropriate level of nurse staffing and to provide a simulation-based education program to support nurses’ clinical judgment. Future studies were proposed to analyze the concept of nursing surveillance by reflecting on the practical experience of clinical nurses and developing a tool that includes multidimensional characteristics of nursing surveillance. The significance of this study is that it investigated the nursing surveillance activities of clinical nurses and identified the influencing factors of the nursing organization in South Korea.

## Figures and Tables

**Table 1 healthcare-11-01273-t001:** Participants’ characteristics (*N* = 205).

Characteristics		*n*	%	M ± SD	Range
Age (years)	<30	128	62.4	29.75 ± 5.02	23.00–52.00
	30–39	64	31.2		
	≥40	13	6.3		
Sex	Male	22	10.7		
	Female	183	89.3		
Total work experience (years)	<3	46	22.4	6.54 ± 5.13	1.00–30.50
	3–4.9	43	21.0		
	5–9.9	83	40.5		
	≥10	33	16.1		
Department	General ward	159	77.6		
	Intensive care unit	46	22.4		
Nurse-to-patient ratio	<1.88	78	38.0	1.88 ± 0.88	0.28–4.24
	≥1.88	127	62.0		
Nursing care delivery method	Functional nursing	54	26.3		
	Team nursing	151	73.7		
Type of hospital	General hospital	103	50.2		
	Tertiary hospital	102	49.8		
Nurses’ perceived expertise	<7.0	70	34.1	6.98 ± 1.43	3.00–10.00
	≥7.0	135	65.9		

Notes. *n*: frequency; M: mean; SD: standard deviation.

**Table 2 healthcare-11-01273-t002:** Descriptive statistics of the variables (*N* = 205).

Variable	Item	M ± SD	Range
Nursing surveillance activity (NSA)	46	149.57 ± 23.39	85–184
Perceived importance of nursing surveillance (PINS)	46	156.94 ± 21.57	92–184
Patient safety culture (PSC) total	33	112.47 ± 15.16	48–165
Teamwork (T)	3	12.20 ± 2.06	1–15
Manpower allocation and workload (MP)	3	8.20 ± 2.27	1–15
Organizational learning and continuous improvement (OL)	3	10.96 ± 1.99	1–15
Error response (ER)	4	11.51 ± 3.17	1–20
Manager support for patient safety (MS)	3	11.87 ± 2.29	1–15
Error communication (EC)	3	11.44 ± 2.77	1–15
Openness in communication (OC)	4	13.94 ± 3.13	1–20
Patient safety incident reporting (PSIR)	2	7.11 ± 1.92	1–8
Hospital management support for patient safety (HMS)	3	9.47 ± 2.55	1–15
Handover-exchange of information (HEI)	3	10.19 ± 2.41	1–15
Overall perceptions of safety (OPS)	1	3.58 ± 0.75	1–5
Experience in patient safety incident reporting (EPSIR)	1	1.97 ± 0.90	1–5

Notes. M: mean; SD: standard deviation.

**Table 3 healthcare-11-01273-t003:** Nursing surveillance activity according to participants’ characteristics (*N* = 205).

Characteristic		M ± SD	*t* or *F* (*p*) (Scheffé Test)
Age (years)	<30	147.98 ± 23.36	1.17 (0.312)
	30–39	151.16 ± 23.98	
	≥40	157.38 ± 19.93	
Sex	Male	144.23 ± 27.21	1.13 (0.258)
	Female	150.21 ± 22.89	
Total work experience (years)	<3	150.96 ± 25.02	1.35 (0.259)
	3–4.9	144.63 ± 22.86	
	5–9.9	149.11 ± 23.50	
	≥10	155.21 ± 20.91	
Department	General wards	148.42 ± 23.86	1.31 (0.193)
	Intensive care unit	153.52 ± 21.44	
Nurse-to-patient ratio	<1.88	154.56 ± 20.55	2.43 (0.016)
	≥1.88	146.50 ± 24.55	
Nursing care delivery method	Functional nursing	147.80 ± 26.54	0.65 (0.518)
	Team nursing	150.20 ± 22.21	
Type of hospital	General hospital	145.42 ± 24.05	2.59 (0.010)
	Tertiary hospital	153.75 ± 22.04	
Nurses’ perceived expertise	<7.0	142.39 ± 25.29	3.24 (0.001)
	≥7.0	153.29 ± 21.51	

Notes. M: mean; SD: standard deviation.

**Table 4 healthcare-11-01273-t004:** Correlation among variables (*N* = 205).

Variable	NSA	PINS	PSC											
			PSC Total	T	MP	OL	ER	MS	EC	OC	PSIR	HMS	HEI	OPS
	*r* (*p*)													
NSA	1													
PINS	0.80 (<0.001)	1												
PSC total	0.26 (<0.001)	0.30 (<0.001)	1											
T	0.22 (0.001)	0.28 (<0.001)	0.65 (<0.001)	1										
MP	−0.02 (0.815)	0.02 (0.748)	0.63 (<0.001)	0.42 (<0.001)	1									
OL	0.29 (<0.001)	0.29 (<0.001)	0.67 (<0.001)	0.55 (<0.001)	0.38 (<0.001)	1								
ER	0.02 (0.774)	0.10 (0.159)	0.66 (<0.001)	0.47 (<0.001)	0.47 (<0.001)	0.38 (<0.001)	1							
MS	0.08 (0.278)	0.14 (0.049)	0.61 (<0.001)	0.55 (<0.001)	0.30 (<0.001)	0.48 (<0.001)	0.41 (<0.001)	1						
EC	0.26 (<0.001)	0.21 (0.003)	0.51 (<0.001)	0.13 (0.067)	0.07 (0.301)	0.23 (0.001)	0.03 (0.686)	0.13 (0.055)	1					
OC	0.28 (<0.001)	0.27 (<0.001)	0.63 (<0.001)	0.18 (0.011)	0.20 (0.004)	0.26 (<0.001)	0.22 (0.002)	0.27 (<0.001)	0.61 (<0.001)	1				
PSIR	0.19 (0.007)	0.19 (0.008)	0.33 (<0.001)	0.07 (0.345)	0.02 (0.805)	0.10 (0.141)	−0.04 (0.608)	0.08 (0.280)	0.38 (<0.001)	0.29 (<0.001)	1			
HMS	0.12 (0.092)	0.11 (0.136)	0.64 (<0.001)	0.27 (<0.001)	0.50 (<0.001)	0.39 (<0.001)	0.45 (<0.001)	0.21 (0.003)	0.21 (0.002)	0.32 (<0.001)	0.11 (0.130)	1		
HEI	0.11 (0.130)	0.15 (0.038)	0.63 (<0.001)	0.51 (<0.001)	0.45 (<0.001)	0.43 (<0.001)	0.49 (<0.001)	0.41 (<0.001)	0.11 (0.113)	0.17 (0.015)	0.01 (0.872)	0.28 (<0.001)	1	
OPS	0.04 (0.566)	0.07 (0.342)	0.12 (0.099)	−0.07 (0.356)	0.07 (0.351)	−0.05 (0.507)	0.04 (0.534)	−0.04 (0.592)	0.08 (0.229)	0.04 (0.527)	0.22 (0.002)	0.12 (0.082)	−0.05 (0.507)	1
EPSIR	0.28 (<0.001)	0.29 (<0.001)	0.41 (<0.001)	0.20 (0.004)	0.22 (0.001)	0.34 (<0.001)	0.15 (0.028)	0.12 (0.088)	0.19 (0.005)	0.25 (<0.001)	0.18 (0.012)	0.33 (<0.001)	0.31 (<0.001)	−0.15 (0.031)

Notes. NSA: Nursing surveillance activity; PINS: Perceived importance of nursing surveillance; PSC: Patient safety culture; T: Teamwork; MP: Manpower allocation and workload; OL: Organizational learning and continuous improvement; ER: Error response; MS: Manager support for patient safety; EC: Error communication; OC: Openness in communication; PSIR: Patient safety incident reporting; HMS: Hospital management support for patient safety; OPS: Overall perceptions of safety; HEI: Handover-exchange of information; EPSIR: Experience in patient safety incident reporting.

**Table 5 healthcare-11-01273-t005:** Factors affecting nursing surveillance activity (*N* = 205).

Variable	Model 1				Model 2				Model 3			
	B	SE	*β*	*t* (*p*)	B	SE	*β*	*t* (*p*)	B	SE	*β*	*t* (*p*)
Constant	138.42	3.10		44.67 (<0.001)	95.59	9.63		9.86 (<0.001)	6.16	8.15		0.76 (0.451)
Nurses’ perceived expertise	10.76	3.32	0.22	3.24 (0.001)	8.77	3.21	0.18	2.73 (0.007)	4.98	2.07	0.10	2.40 (0.017)
Tertiary hospital *	8.16	3.15	0.18	2.59 (0.010)	5.50	3.06	0.12	1.80 (0.074)	0.68	1.98	0.15	0.34 (0.731)
PSC												
OL					2.23	0.80	0.19	2.79 (0.006)	0.43	0.52	0.04	0.83 (0.409)
OC					1.51	0.50	0.20	3.00 (0.003)	0.41	0.33	0.06	1.26 (0.209)
PINS									0.82	0.05	0.76	16.96 (<0.001)
F (*p*)	8.75 (<0.001)				10.33 (<0.001)				77.65 (<0.001)			
Adjusted R^2^	0.071				0.155				0.653			
R^2^ change					0.091				0.490			
Tolerance	1.000				0.895–0.971				0.849–0.960			
VIF	1.000				1.029–1.117				1.042–1.178			
Durbin-Watson	2.065				1.966				1.970			

Notes. B: standardized regression coefficient; SE: standard error; *β*: standardized regression coefficient; VIF: variance inflation factor; PSC: Patient safety culture; OL: Organizational learning and continuous improvement; OC: Openness in communication; PINS: Perceived importance of nursing surveillance. Model 1 factors: Characteristics of participants; Model 2 factors: patient safety culture; Model 3 factors: perceived significance of nursing surveillance. * Dummy variable: Type of hospital (ref. = general hospital).

## Data Availability

Not applicable.

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
