# Peer review of "Factors Affecting Nursing Surveillance Activity among Clinical Nurses"

_healthcare, 2023, doi:10.3390/healthcare11091273_

Round 1

Reviewer 1 Report

The authors have presented an interesting and important study that has significant implications for patient safety. I wonder since the dependent variable measured in this study (surveillance) is an ordinal variable- should the authors have performed an ordinal regression analysis? 

Author Response

Response to Reviewer 1 Comments

Comments and Suggestions for Authors

The authors have presented an interesting and important study that has significant implications for patient safety.

Authors’ Response: We appreciate the time and effort that you and the reviewers have put into providing valuable feedback and insightful comments, which improved our manuscript. We have carefully considered each comment and updated the manuscript, as required. We have marked the revisions made to the manuscript in red font.

Point 1: I wonder since the dependent variable measured in this study (surveillance) is an ordinal variable- should the authors have performed an ordinal regression analysis?

Response 1: Most of the tools used in nursing are composed of items on a 5-point Likert scale. Strictly speaking, the Likert scale is ordinal, but we have analyzed it as a quantitative variable rather than qualitative by making it an interval scale. Stevens (1946), who created the scale, also mentioned that if the number of Likert scales is increased, it is possible to perform arithmetic operations on a continuous scale.

Huiping Wu & Shing-On Leung (2017) Can Likert Scales be Treated as Interval Scales?-A Simulation Study, Journal of Social Service Research, 43:4, 527-532, DOI: 10.1080/01488376.2017.1329775

The following is an explanation of the tools in the study of Kelly (2009) who developed a nursing surveillance tool.

Respondents were asked to evaluate each item and rate the frequency on a 1-5 Likert scale ranging from “Never” to “Always”. ~ This allowed for a single number response that represented the frequency ~ of the activity. This method was preferred to a summed total scale to have an interpretable continuum scale of activities.

Kelly, L.A. Nursing Surveillance in the Acute Care Setting: Latent Variable Development and Analysis. The University of Arizona, 2009.

Reviewer 2 Report

It was my pleasure to review your article on Factors Affecting Nursing Surveillance Activity Among Clinical Nurses. I found the article to be timely as it points out the important role of nurses in patient safety and advances the work related to the role of nurses in patient safety. Like so many areas of nursing COVID-19 intensified the need to address the role of nurses and the need for surveillance.

The background builds on the abstract and clearly identifies gaps in the literature. The study methods and participant selection were well-designed and explained. If there was a theory or conceptual framework on which the study was based, it is not clear. It would be nice to see theory addressed in the methods or to explain that no specific theory was used, but rather, the concepts of surveillance are addressed.

Recommendations:

Line 41: It is unclear what is meant by surveillance as a defense mechanism since a defense mechanism is generally considered unconscious and avoids conflict or decreases internal stress. It is only in the context of the Eindhoven Model that it is clear. It may be helpful to add a sentence about the Eindhovern model from the article that was cited. Specifically, addressing that the model is referring to surveillance as a defense mechanism that “blocks dangerous situations.”

Line 30, 32, and 70 there are 4 slightly different definitions or conceptualizations of surveillance. It may be better to either put the various definitions together or, in the last paragraph, make a statement about the definition/conceptualization utilized in this study. In other words, provide your own definition that guides your study.

Conclusion: I would suggest restating some of the primary recommendations be included in the conclusion. While they are well explained in the discussion, it may be good to highlight the significant recommendations again as a summary.

Author Response

Response to Reviewer 2 Comments

Comments and Suggestions for Authors

It was my pleasure to review your article on Factors Affecting Nursing Surveillance Activity Among Clinical Nurses. I found the article to be timely as it points out the important role of nurses in patient safety and advances the work related to the role of nurses in patient safety. Like so many areas of nursing COVID-19 intensified the need to address the role of nurses and the need for surveillance.

Authors’ Response: We appreciate the time and effort that you and the reviewers have put into providing valuable feedback and insightful comments, which considerably improved our manuscript. We have carefully considered each comment and made changes to the manuscript, as required. We have marked in red font the revisions we made to the manuscript.

Point 1: The background builds on the abstract and clearly identifies gaps in the literature. The study methods and participant selection were well-designed and explained. If there was a theory or conceptual framework on which the study was based, it is not clear. It would be nice to see theory addressed in the methods or to explain that no specific theory was used, but rather, the concepts of surveillance are addressed.

Response 1: We would like to express our deepest gratitude for your review of our study. Unfortunately, as pointed out, only few theories have explained nursing surveillance, and concept analysis studies of nursing surveillance that have been reported so far have been described in lines 71–73.

Point 2: Line 41: It is unclear what is meant by surveillance as a defense mechanism since a defense mechanism is generally considered unconscious and avoids conflict or decreases internal stress. It is only in the context of the Eindhoven Model that it is clear. It may be helpful to add a sentence about the Eindhoven model from the article that was cited. Specifically, addressing that the model is referring to surveillance as a defense mechanism that “blocks dangerous situations.”

Response 2: We appreciate your critical comment. We have added a reference and a sentence about the Eindhoven model as follows.

“Nursing surveillance is a defense mechanism that protects patients from adverse events according to the Eindhoven Model [7] and systematically contributes to nurses’ detection of and intervention in dangerous situations [8].” (lines 44–46)

  1. Van der Schaff TW. Near-Miss Reporting in the Chemical Process Industry [thesis]. Eindhoven, The Netherlands: Eindhoven University of Technology; 1992. (lines 406–407)

Point 3: Line 30, 32, and 70 there are 4 slightly different definitions or conceptualizations of surveillance. It may be better to either put the various definitions together or, in the last paragraph, make a statement about the definition/conceptualization utilized in this study. In other words, provide your own definition that guides your study.

Response 3: Thank you for your thoughtful comment. We have provided the definition that guides our study.

“In this study, nursing surveillance uses the definition of purposeful and ongoing acquisition, interpretation, and synthesis of patient data for clinical decision-making, and nursing surveillance activity was measured by the activities of surveillance in NIC [4].” (lines 86–89)

Point 4: Conclusion: I would suggest restating some of the primary recommendations be included in the conclusion. While they are well explained in the discussion, it may be good to highlight the significant recommendations again as a summary.

Response 4: Thank you for your valuable feedback. We have summarized the results and added important recommendations in the conclusion.

  1. Conclusions

Nursing surveillance around the clock is crucial to patients’ health and safety. This study found that nurses’ characteristics and work environment can be proposed as antecedents of nursing surveillance [3,6,8,39], specifically, nurses’ expertise, their perception of nursing surveillance, patient safety culture, and type of hospital-predicted nursing surveillance activity. In order to promote nurses' surveillance activities in acute hospitals, it is necessary to maintain an appropriate level of nurse staffing and to provide a simulation-based education program to support nurses' clinical judgment. Future studies were proposed that analyze the concept of nursing surveillance by reflecting on the practical experience of clinical nurses and developing a tool that includes multidimensional characteristics of nursing surveillance. The significance of this study investigated the nursing surveillance activities of clinical nurses and identified the influencing factors of the nursing organization in South Korea. (lines 366–377)

Reviewer 3 Report

Thank you for the opportunity to review your manuscript titled Factors Affecting Nursing Surveillance Activity Among Clinical Nurses. It's an interesting manuscript that try to identify the predictors of nursing surveillance activity. I have somme comments for authors:

Keywords: Please, use MeSH terms. MeSH terms will help index your publication in databases. E.g: one MeSH term of your study would be safety management.

Author Response

Response to Reviewer 3 Comments

Comments and Suggestions for Authors

Thank you for the opportunity to review your manuscript titled Factors Affecting Nursing Surveillance Activity Among Clinical Nurses. It's an interesting manuscript that try to identify the predictors of nursing surveillance activity. I have some comments for authors:

Authors’ Response: We appreciate the time and effort that you and the reviewers have put into providing valuable feedback and insightful comments, which improved our manuscript. We have carefully considered each comment and updated the manuscript, as required. We have marked the revisions made to the manuscript in red font.

Point 1: Keywords: Please, use MeSH terms. MeSH terms will help index your publication in databases. E.g: one MeSH term of your study would be safety management.

Response 1: Thank you for your valuable feedback. We have changed the keywords to MeSH terms.

Keywords: safety management; nursing; patient safety; professional competence (line 25)

Reviewer 4 Report

Thank you to the journal for having me as a reviewer. Dear authors, I am aware of the effort it takes to conduct research. My intention is to help you to improve your manuscript.

The research assesses the activity of nursing surveillance, the perceived importance thereof, as well as patient safety culture in an attempt to identify predictive factors.

The study is relevant and although it uses the Nursing Interventions Classification, external validity with respect to other countries is not possible. This drawback encourages repeating the study in other countries, cultures, and nursing institutions.

Categories with a low number of participants have been noted by the authors.

Minor corrections are required:

Please revise the entire text to unify the presentation format of numbers (leading zero before decimal point).

Convert the tables to the correct format: separate variables according to categories, align the notes in Table 4, and visually order to facilitate reading.

Remove the double dash in the reference on line 239.

Remove the dash in the references on line 363.

55% of the references are more than ten years old. It is recommended to use updated references and leave the older ones for classic references or in topics where there is no update available.

Author Response

Response to Reviewer 4 Comments

Comments and Suggestions for Authors

Thank you to the journal for having me as a reviewer. Dear authors, I am aware of the effort it takes to conduct research. My intention is to help you to improve your manuscript. The research assesses the activity of nursing surveillance, the perceived importance thereof, as well as patient safety culture in an attempt to identify predictive factors.

The study is relevant and although it uses the Nursing Interventions Classification, external validity with respect to other countries is not possible. This drawback encourages repeating the study in other countries, cultures, and nursing institutions.

Categories with a low number of participants have been noted by the authors.

Authors’ Response: We appreciate the time and effort that you and the reviewers have put into providing valuable feedback and insightful comments, which improved our manuscript. We have carefully considered each comment and updated the manuscript, as required.

We have marked the revisions made to the manuscript in red font.

Point 1: Please revise the entire text to unify the presentation format of numbers (leading zero before decimal point).

Response 1: We have unified and corrected the text by deleting 0 in front of the decimal point for values less than 1.

Point 2: Convert the tables to the correct format: separate variables according to categories, align the notes in Table 4, and visually order to facilitate reading.

Response 2: Thank you for your valuable feedback. The format of the tables was modified and variables were separated according to categories. Particularly, for patient safety culture (PSC), total and subdomain variables were listed again in Table 2 to make it easier for readers to understand when reading Table 4. The notes in Table 4 were arranged in the order in which the variables appeared, and the missing variable names were added to the notes.

Point 3: Remove the double dash in the reference on line 239.  

Response 3: We have removed the double dash in the reference on line 239.

A hierarchical regression analysis was performed to identify the predictors of nursing surveillance activity; the regression model is presented in Table 5. The categorical variables that were significant in the univariate analysis such as nurse-to-patient ratio and type of hospital were dummy-coded, and the nurse’s perceived expertise was entered as a continuous variable. (lines 241–245)

Point 4: Remove the dash in the references on line 363.

Response 4: We have removed the double dash in the reference on line 363 and changed references 7 and 8.

Nursing surveillance around the clock is crucial to patients’ health and safety. This study found that nurses’ characteristics and work environment can be proposed as antecedents of nursing surveillance [3,6,8,39], specifically, nurses’ expertise, their perception of nursing surveillance, patient safety culture, and type of hospital-predicted nursing surveillance activity. (lines 366–370)

Point 5: 55% of the references are more than ten years old. It is recommended to use updated references and leave the older ones for classic references or in topics where there is no update available.

Response 5: We have reviewed the references and replaced them with the latest references wherever possible.

Reviewer 5 Report

In the research miss inclusion and exclusion criteria for respondents

Author Response

Response to Reviewer 5 Comments

Comments and Suggestions for Authors

In the research miss inclusion and exclusion criteria for respondents.

Authors’ Response: We appreciate the time and effort that you and the reviewers have put into providing valuable feedback and insightful comments, which improved our manuscript. We have carefully considered each comment and updated the manuscript, as required. We have marked the revisions made to the manuscript in red font.

Point 1: In the research miss inclusion and exclusion criteria for respondents.

Response 1: Thank you for your thoughtful comment. The inclusion criteria were nurses with at least one year of clinical experience recruited from two secondary hospitals and two tertiary hospitals in Changwon-si and Cheongju-si, South Korea, via a voluntary online survey that explained the study purpose and methods. This was described in the Participants subsection of the Methods section. Subjects of this study were excluded if they had less than 1 year of work experience, worked at a general hospital, or were not currently working as nurses.

2.2. Participants

A total of 205 nurses with at least one year of clinical experience were recruited from two secondary hospitals and two tertiary hospitals in Changwon-si and Cheongju-si, South Korea, via a voluntary online survey that explained the study purpose and methods. (lines 98–101)

Reviewer 6 Report

This is a very well written research paper that follows the research design effectively. Very good references have been used to support the data as well. Only minor spell check is recommended from me at this time, great work overall.

1. What is the main question addressed by the research?   

What is the degree of nursing surveillance activity performed by clinical nurses and what is the importance of nursing surveillance and nurses perception of how nursing surveilliance attributes to a culture of patient safety. 
2. Do you consider the topic original or relevant in the field? Does it
address a specific gap in the field?   Yes, i consider the topic relevant in the field as it relates to improving the safety of healthcare. It address a patient safety and quality of care gaps in healthcare. 
3. What does it add to the subject area compared with other published
material?  This study adds very relevant data from a large sample size (in South Korea) providing a strong assessment of the information researched
4. What specific improvements should the authors consider regarding the
methodology? What further controls should be considered?  Another hospital could have been added to create a larger sample size.  Another hospital in that area could have beeen used as a control, or nurses of varying backgrounds and experiences.
5. Are the conclusions consistent with the evidence and arguments presented
and do they address the main question posed? Yes the conclusions address the main questions presented 
6. Are the references appropriate? yes, the references were all relevant and supported the research conducted 
7. Please include any additional comments on the tables and figures.  the table was good but could have been simplified 

Author Response

Response to Reviewer 6 Comments

Comments and Suggestions for Authors

This is a very well written research paper that follows the research design effectively. Very good references have been used to support the data as well. Only minor spell check is recommended from me at this time, great work overall.

Author’s Response: We appreciate the time and effort that you and the reviewers have put into providing valuable feedback and insightful comments, which improved our manuscript. We have carefully considered each comment and updated the manuscript, as required.

We have marked the revisions made to the manuscript in red font.

Point 1: 1. What is the main question addressed by the research? What is the degree of nursing surveillance activity performed by clinical nurses and what is the importance of nursing surveillance and nurses perception of how nursing surveilliance attributes to a culture of patient safety.

  1. Do you consider the topic original or relevant in the field? Does it address a specific gap in the field? Yes, i consider the topic relevant in the field as it relates to improving the safety of healthcare. It address a patient safety and quality of care gaps in healthcare.
  2. What does it add to the subject area compared with other published material? This study adds very relevant data from a large sample size (in South Korea) providing a strong assessment of the information researched.
  3. What specific improvements should the authors consider regarding the methodology? What further controls should be considered? Another hospital could have been added to create a larger sample size. Another hospital in that area could have beeen used as a control, or nurses of varying backgrounds and experiences.
  4. Are the conclusions consistent with the evidence and arguments presented and do they address the main question posed? Yes the conclusions address the main questions presented
  5. Are the references appropriate? yes, the references were all relevant and supported the research conducted

Response 1: Thank you very much for your valuable feedback.

Point 2: Only minor spell check is recommended from me at this time.

Response 2: We did a meticulous spell check in this revision and then corrected it. Since English is not our native language, we availed of the editing service of a professional editing company (Editage.co.kr). We have attached the editing certificate of the editing service.

Point 3: Please include any additional comments on the tables and figures. The table was good but could have been simplified.

Response 3: Thank you for your valuable feedback. Table 2 was simplified by deleting the variable's standardized scale scores. Although the sub-domain of patient safety culture was somewhat complicated, please understand that it was left as it was in order to identify the factors that affected nursing surveillance in this study.

Reviewer 7 Report

The manuscript entitled “Factors Affecting Nursing Surveillance Activity Among Clinical Nurses” explored the predictors of nursing surveillance activity, promoting an assessment of the extent and key predictors of nursing surveillance activity. However, the workload of this study is too small and the writing is illogical. In addition, there are some other problems that need to be fully addressed for this manuscript to be considered for publication.

1. The ABSTRACT should be improved by briefly supplementing the results of this study to make readers can understand the whole research only by reading the ABSTRACT.

2. The INTRODUCTION was badly written. The background is lack logic and looks like an accumulation of a bunch of papers. Moreover, the background took much space, and the methods and significance of this research were a lack in the INTRODUCTION. Therefore, the authors are suggested to rewrite the whole INTRODUCTION, including reducing the content of the background and rewriting it more logically, supplementing a brief description of the method and significance of this research.

3. It would be better if the authors could provide the Hospital Survey on Patient Safety Culture TM 140 (SOPSTM) Version 2.0 and other tools used for statistics as supplement files. 

4. Table 1: The underlining of “Age (years)” needs to be declined.

Table 3: The underlining of “Age (years)” needs to be declined.

Table 4: The underlining of “NSA” and “1” is not necessary.

Table 5 is hard to read and should be improved.

5. It would be better if the authors could supplement the advantages, limitations, and future perspective of this research in the CONCLUSION part.

6. The manuscript contained a few grammatical errors and would benefit from careful editing. This study and other studies cited in this study have been finished, so it would be better to use the past tense to discuss their results. The authors are suggested to check the manuscript carefully to avoid these mistakes.

Author Response

Response to Reviewer 7 Comments

Comments and Suggestions for Authors

The manuscript entitled “Factors Affecting Nursing Surveillance Activity Among Clinical Nurses” explored the predictors of nursing surveillance activity, promoting an assessment of the extent and key predictors of nursing surveillance activity. However, the workload of this study is too small and the writing is illogical. In addition, there are some other problems that need to be fully addressed for this manuscript to be considered for publication.

Authors’ Response: We appreciate the time and effort that you and the reviewers have put into providing valuable feedback and insightful comments, which improved our manuscript. We have carefully considered each comment and updated the manuscript, as required. We have marked the revisions made to the manuscript in red font.

Point 1: The ABSTRACT should be improved by briefly supplementing the results of this study to make readers can understand the whole research only by reading the ABSTRACT.

Response 1: We appreciate your critical comments. We have added a results section in the abstract.

“Nursing surveillance activity was high among nurses assigned fewer than 1.88 patients, who worked in a tertiary hospital, and those who scored 7.0 or higher in nurses’ perceived expertise. Nursing surveillance activity was significantly positively correlated with nurses’ perceived importance of nursing surveillance and patient safety culture. A hierarchical multiple regression analysis was performed to identify the predictors of nursing surveillance activity. Nurses’ perceived expertise, perceived importance of nursing surveillance, patient safety culture, and type of hospital explained 65.3% of the variance of nursing surveillance activity. This study holds significance as it provides an assessment of the extent and key predictors of nursing surveillance activity.” (lines 17–24)

Point 2: The INTRODUCTION was badly written. The background is lack logic and looks like an accumulation of a bunch of papers. Moreover, the background took much space, and the methods and significance of this research were a lack in the INTRODUCTION. Therefore, the authors are suggested to rewrite the whole INTRODUCTION, including reducing the content of the background and rewriting it more logically, supplementing a brief description of the method and significance of this research.

Response 2: Thank you for your valuable feedback. Research on nursing surveillance was in the stage of analyzing concepts and classifying sub-domains of concepts. We have thoroughly reviewed previous research. The necessity for research to identify the effects of patient safety culture and perception of the importance of nursing surveillance on nursing surveillance activities was logically described. As we pass through the COVID-19 pandemic, nursing surveillance was a more important part of nursing care; hence, we will not be revising these logical descriptions. We apologize for not fully accepting your comments.

Point 3: It would be better if the authors could provide the Hospital Survey on Patient Safety Culture TM 140 (SOPSTM) Version 2.0 and other tools used for statistics as supplement files.

Response 3: The tool is available in several national languages on the AHRQ website:

https://www.ahrq.gov/sops/surveys/hospital/index.html. We have added the URL of the website where the questionnaire can be obtained as follows.

2.3.2. Patient safety culture

The Korean version [33] of the Hospital Survey on Patient Safety Culture TM (SOPSTM) Version 2.0, originally developed by the US Agency for Healthcare Research and Quality (https://www.ahrq.gov/sops/surveys/hospital/index.html), was used after obtaining permission from the authors.

Point 4: Table 1: The underlining of “Age (years)” needs to be declined.

Response 4: We have deleted the underlining of “Age (years)” in Table 1.

Point 5: Table 3: The underlining of “Age (years)” needs to be declined.

Response 5: We have deleted the underlining of “Age (years) in Table 3.

Point 6: Table 4: The underlining of “NSA” and “1” is not necessary.

Response 6: We have deleted the underlining of “NSA” and “1” in Table 4.

Point 7: Table 5 is hard to read and should be improved.

Response 7: We are really sorry for the inconvenience caused. We have organized the variable columns and abbreviated and explained variable names as NOTES in Table 5 to make it easier for readers to understand.

Point 8: It would be better if the authors could supplement the advantages, limitations, and future perspective of this research in the CONCLUSION part.

Response 8: We appreciate your thoughtful comment. The limitations of the study and research suggestions were described in the last paragraphs of the discussion, and the main results, strategies, and significance of the study were described in the Conclusion.

  1. Conclusions

Nursing surveillance around the clock is crucial to patients’ health and safety. This study found that nurses’ characteristics and work environment can be proposed as antecedents of nursing surveillance [3,6,8,39], specifically, nurses’ expertise, their perception of nursing surveillance, patient safety culture, and type of hospital-predicted nursing surveillance activity. In order to promote nurses' surveillance activities in acute hospitals, it is necessary to maintain an appropriate level of nurse staffing and to provide a simulation-based education program to support nurses' clinical judgment. Future studies were proposed that analyze the concept of nursing surveillance by reflecting on the practical experience of clinical nurses and developing a tool that includes multidimensional characteristics of nursing surveillance. The significance of this study investigated the nursing surveillance activities of clinical nurses and identified the influencing factors of the nursing organization in South Korea. (lines 366–377)

Point 9: The manuscript contained a few grammatical errors and would benefit from careful editing. This study and other studies cited in this study have been finished, so it would be better to use the past tense to discuss their results. The authors are suggested to check the manuscript carefully to avoid these mistakes.

Response 9: Thank you for your valuable feedback. We have carefully reviewed the revised manuscript, and since we do not speak English as our native language, we have availed of the editing service of a professional editing company (Editage.co.kr). We have attached the editing certificate of the editing service.

Reviewer 8 Report

Thank you for the opportunity to participate in the review of this manuscript.

The manuscript with the title "Factors Affecting Nursing Surveillance Activity Among Clinical Nurses" has a particular importance for the activity of nurses, because through the conducted research the main factors that can influence nursing activities are analyzed, considering the need at a global level to ensure a high standard of medical care and ensuring patient safety.

The study is well designed and follows the structure of a descriptive cross-sectional study.

We did not identify a process of excessive self-citation.

I have a question regarding the methodology section:

- What were the measures to limit the phenomenon of bias?

In the "discussions" section, I recommend to  discuss limitations of the study, taking into account sources of potential bias or imprecision. Discuss both direction and magnitude of any potential bias.

These were my recommendations for improving the manuscript.

I congratulate the authors once again for the work done.

Author Response

Response to Reviewer 8 Comments

Comments and Suggestions for Authors

Thank you for the opportunity to participate in the review of this manuscript. The manuscript with the title "Factors Affecting Nursing Surveillance Activity Among Clinical Nurses" has a particular importance for the activity of nurses, because through the conducted research the main factors that can influence nursing activities are analyzed, considering the need at a global level to ensure a high standard of medical care and ensuring patient safety.

The study is well designed and follows the structure of a descriptive cross-sectional study. We did not identify a process of excessive self-citation. These were my recommendations for improving the manuscript. I congratulate the authors once again for the work done.

Authors’ Response: We appreciate the time and effort that you and the reviewers have put into providing valuable feedback and insightful comments, which considerably improved our manuscript. We have carefully considered each comment and made changes to the manuscript, as required. We have marked in red font the revisions we made to the manuscript.

Point 1: I have a question regarding the methodology section: What were the measures to limit the phenomenon of bias? In the "discussions" section, I recommend to discuss limitations of the study, taking into account sources of potential bias or imprecision. Discuss both direction and magnitude of any potential bias.

Response 1: We would like to express our deepest gratitude for your review of our study. As a cross-sectional survey, this study surveyed nurses at two general hospitals and two tertiary hospitals in two cities to reduce bias in the sampling process. Nevertheless, it was additionally described that there is a limitation in the generalization of the results due to the convenience sampling method at the end of the Discussion section.

The cross-sectional nature of this study hinders a causal analysis among the variables, and this study has limitations in generalizing the results because the survey was conducted on nurses from two general hospitals and two tertiary hospitals in two cities in South Korea selected by convenience sampling. In the future, studies should conduct multi-center studies to investigate nursing surveillance activity among different types of nursing delivery systems and analyze the associations of nursing surveillance with the factors proposed as outcomes of nursing surveillance. Additionally, given that the nursing surveillance interventions listed on the NIC were used in this study, we recommend that subsequent studies conceptualize nursing surveillance from clinical nurses’ perspectives and explore their lived experiences to develop and validate instruments that capture the multidimensional nature of nursing surveillance.
